# Anti-Diabetic and Cytotoxic Evaluation of *Phlomis stewartii* Plant Phytochemicals on Cigarette Smoke Inhalation and Alloxan-Induced Diabetes in Wistar Rats

**DOI:** 10.3390/metabo12111133

**Published:** 2022-11-17

**Authors:** Mamoon Ur Rasheed, Syed Ali Raza Naqvi, Nasir Rasool, Syed Adnan Ali Shah, Zainul Amiruddin Zakaria

**Affiliations:** 1Department of Chemistry, Government College University, Faisalabad 38040, Pakistan; 2Faculty of Pharmacy, Universiti Teknologi MARA Cawangan Selangor Kampus Puncak Alam, Bandar Puncak Alam 42300, Selangor, Malaysia; 3Atta-ur-Rahman Institute for Natural Product Discovery (AuRIns), Universiti Teknologi MARA Cawangan Selangor Kampus Puncak Alam, Bandar Puncak Alam 42300, Selangor, Malaysia; 4Malaysia Borneo Research on Algesia, Inflammation and Neurodegeneration (BRAIN) Group, Faculty of Medicine and Health Sciences, Sabah Universiti Malaysia, Jalan UMS, Kota Kinabalu 88400, Sabah, Malaysia

**Keywords:** inflammation, antidiabetic, oxidative stress, metabolic syndrome, *P. stewartii* Hf., diabetes mellitus

## Abstract

The generation of free radicals in body causes oxidative stress and consequently different metabolic disorders. There are numerous environmental and emotional factors that trigger free radical generation, cigarette smoke (CS) is one of them. In addition to free radical production, it also increases the risk of developing type II diabetes, cancer, and has adverse effects on other organs such as liver and kidneys. In the present study, extracts of leaves, flower, and whole plant of *P. stewartii* Hf. in methanol were analyzed using LC-ESI-MS and investigated for their cytotoxic properties against HepG2 cell line and CS alloxan-induced diabetes in Wistar albino rats model. A total of 24 rats were kept in aerated cage for eight weeks and exposed to CS following the administration of single dose of alloxan@140 mg/kg body weight at the end of six weeks to induce diabetes mellitus (DM). The cytotoxic activity of extracts against HepG2 was recorded in the order; leaves methanol (LM) > flower methanol (FM) and whole plant methanol (WPM). The IC50(1/4) values were in the order of 187 (LM) > 280 (FM) > 312 (WPM) µg/mL against HepG2. In positive control group, CS- and alloxan-induced diabetes significantly increased (*p* < 0.05) the level of alanine alkaline phosphatase (ALP), aminotransferase (ALT), aspartate aminotransferase (AST), low density lipoprotein (LDL), bilirubin, total protein, creatinine, uric acid, blood urea, globulin, total oxidant status (TOS), and malondialdehyde (MDA), as compared to negative control group. In conclusion, according to the results of this study, *P. Stewartii* methanol extracts showed good antioxidant, anticancer activity and worked well to recover the tested clinical parameters in CS/alloxan-induced diabetes animals, which indicated the extracts also possess good antidiabetic, hepatoprotective, and nephroprotective potential.

## 1. Introduction

Living organism produces reactive oxidative species (ROS) through endogenous sources as part of cellular metabolism and exogenous sources including CS [1]. ROS damages the cell structure such as lipids, carbohydrates, proteins, nucleic acids and distort their metabolic functions as well [2]. The imbalanced shift in production of oxidants against antioxidants is termed as oxidative stress (OS) [3,4]. In addition to numerous environmental and emotional factors, cigarette smoke (CS) also triggers the production of ROS. CS having more than 400 different chemicals and oxidants results in the generation of ROS and has adverse impact on body tissues [5]. In addition, CS generate ROS by stimulating the NADPH oxidase (Noxs) which activates the inflammatory cells, increases polymorphonuclear cells in different tissues of body such as liver and kidney which mediate the athogenesis for metabolic diseases [6,7]. Liver diseases are the major cause of disability and death and have become a global health concern [8]. Acute and chronic inflammation in liver injuries develop cirrhosis that results in steatohepatitis and ultimately hepatocellular carcinoma (HCC) [9].

Alloxan, a synthetic drug which is produced by the oxidation of uric acid, potentially causes the necrosis of the pancreas and also triggers ROS [10]. The diabetogenicity of alloxan is due to generation of ROS. Alloxan exhibits high affinity toward sulfhydryl (SH) group containing cellular components which cause the reduction of glutathione (GSH), cysteine, ascorbate and inhibits glucokinase activity. The inhibition of glucokinase suppresses the glucose oxidation and adenosine triphosphate (ATPs) production which hampers insulin secretion [10,11]. Alloxan binds with two sugar-binding sites of (SH) group and forms a disulfide bond and causes inactivation of GSH, cysteine, ascorbate, and other enzymes [12,13].

DM is a metabolic and endocrine malady which occurs through hyperglycemia and irregular functions of β cells of pancreatic tissue [14]. CS increases the risk of developing insulin-dependent DM (IDDM) by 40–50% which makes it challenging for peoples to control carbohydrates and lipid metabolism [15]. However, CS does not directly affect the liver and kidneys but it is an additional source of free radicals and shifts the balance of oxidants and antioxidant status toward more oxidant production in the body [16]. World Health Organization (WHO) reported, annually about 2 billion people are affected directly by inhalation of CS [17]. Traditionally, a variety of plant extracts are used to nullify the adverse effects of CS/OS induced disorders, typically to treat asthma, cough and respiratory tract infections, and DM [18,19,20]. Metabolites present in plants such as steroid, resins, phenol, alkaloids, flavonoids, gum, and tannins possess antibacterial, antiviral, antifungal, an tioxidant, antimicrobial, and anticancer properties [21]. Prospectively, with the ever increasing population and emergence of microbial resistance to existed synthetic drugs, a strong assistance will be felt to address the viral, microbial, and metabolic diseases [18]. *Phlomis stewartii* belongs to the lamiaceae family. The plant was located in Pakistan and was first collected by Stewart, J.L. and latter it was assigned a botanical name, *Phlomis stewartii* Hook.f. (*P. stewartii* Hf.) and specimen barcode/id (K000894367) [22]. It grows during the month of June–August. It has already been reported that lamiaceae family bears potent medicinal phytochemicals that are being used as an antiseptic, anthelmintic, and antidiabetic herbal medicines [23]. Previously, it was reported that α-glucosidase inhibitors such as caffeic acids, p-hydroxybenzoic acid, phenylethanoid glycoside, and notohamosin were isolated from *P. stewartii* Hf. extract which indicate that the methanol extracts of different parts of plant could be tested as antidiabetic, hepatoprotective, and nephroprotective products [24]. The aim of our study was to examine the medicinal potential of *P. stewartii* Hf. plant extracts in term of cytotoxic potential, serological, and histopathological analysis of liver and kidney tissues using CS and alloxan induced diabetes in rat models.

## 2. Materials and Methods

### 2.1. Collection and Identification of Plant

*P. stewartii* Hf. was selected to explore its medicinal importance. Plants were collected from the desert area of Baluchistan during the month of June–August in 2018. The plant was authenticated by Department of Botany, University of Balochistan, Quetta Pakistan.

### 2.2. Washing and Separation of Plant Parts

Leaves and flowers were separated from *P. stewartii* Hf. plant, and washed with distilled water and dried at normal room temperature (30 °C) for more than two weeks to decrease the moisture contents. Then, leaves, flowers, and whole plant parts were pulverized by using the mechanical blender to convert into powdered form. Special air tight polythene bags were used with proper labeling to store the fine powder for further analysis.

### 2.3. Washing and Separation of Plant Parts

Extraction of bioactive constituents was carried out by soaking 10 g plant powder in 200 mL methanol, followed by stirring mechanically at 150 rpm for 8 h by using an orbital shaker. Whatman No.1 filter paper was used for filtering the extract mixture. The filtrate then subjected to rotary evaporator to get semi solid extract at 32 °C. Semi solid samples were kept in the refrigerator at 4 °C for further analysis.

### 2.4. LC-ESI-MS Analysis

The phenolics in P. stewartii Hf. extracts were analyzed by liquid chromatography combined with liquid chromatography–electrospray ionization mass spectrometry (LC-ESI-MS). The membrane of 0.45 µm was used to filter the plant extracts. The separation of phenolic components was performed on a surveyor Plus HPLC system equipped with Auto Surveyor (Thermo Scientific, San Joie, CA, USA). Pump was connected with RP C18 analytical column (4.6 × 150 mm) bearing 3.0 µm particle size of stationary phase. The elution solvent consists of LC-MS grade methanol and acidified water (0.5% formic acid *v*/*v*) as mobile phase A and B, respectively. The solvent elution at a flow rate of 0.3 mL/min was performed in a gradient system. Moreover, the gradient elution was programmed as follows: 10% to 30% A and 90 to 70% B from 0 to 10 min followed by 30 to 50% A and 70 to 50% B in next 20 min and this flow was maintained till the end of the investigation. The 2 min re-equilibration time was used after each analysis. At 25 °C the column was maintained and the injection volume was 5.0 µL. Effluent from HPLC was directed to the electron spray ionization mass spectrometer 9LTQXLTM linear ion trap Thermo Scientific, (River Oaks Parkway, San Jose, CA, USA). The mass spectrometer was also equipped with an ESI ionization source. For analysis of different parameters were set by using negative ion mode with spectra acquired over a mass in domain of *m*/*z* 260 to 800. Optimum range of ESI-MS parameters were as given: spray voltage, +4.0 KV: auxiliary gas and sheath gas 5 and 45 units/min, respectively; tube lens voltage −66.51 V, capillary voltage, −20.0 V, capillary temperature 320 °C by using Excalibur Software (Thermo Fisher Scientific Inc, Waltham, MA, USA).

### 2.5. Determination of Cytotoxicity

The human hepatocellular carcinoma cell line (HepG2) was maintained according to the guidelines given by ATCC. HepG2 cell was cultured in DMEM with 0.1% streptomycin/penicillin and 10% of fetal bovine serum. These cells were incubated under an atmosphere containing 5% CO_2_ in a humidified incubator. The cells were sub-cultured at preconfluent densities by using 0.25% trypsin-EDTA. The cytotoxicity potential of methanol extracts of *P. stewartii* Hf. plant on cultured cell was measured using the MMT assay [25]. To proceed, 96-well plates were used for growth of cells at a density of 5 × 10^4^ cells per plate. After the incubation of 24 h, cells were treated with various concentration of samples and incubated for 48 h. Then MTT solution (25 μL of 5 mg/mL, Roche) was added to each well and all plates were re-incubated for 4 h. The medium was removed after addition of 100 µL of DMSO to solubilize the formed formazan crystal. The total amount of formazan crystal was observed by measuring the absorbance at 940 nm using a microplate spectrophotometer. The 50% cytotoxic concentration (IC_50_) of all constituents was calculated and all assays were performed in triplicates.

### 2.6. Cigarette Smoke Inhalation and Alloxan-Induced Diabetes

Rats were introduced to CS chamber with (dimensions: 1.000 × 850 × 750 mm^3^) attached suction pump in accordance to previous studies. CS was drawn out from cigarette (composition per unit: 14 mg tar, 1.1 mg of nicotine and 15 mg carbon monoxide) by using suction and filled the enclosed chamber. Number of cigarette was gradually increased in the first week from 08 to 20 for 1 h, and the rats were exposed twice a day. Consequently, in each smoking chamber 20 cigarettes were used (1 h in morning and 1 h in afternoon session) in 120–130 g rats for eight weeks. When rats expressed cherry red to purple plantar skin, mouth breathing, agitation, mouth wheezing, or upon the 3 min time limit, the CS was stopped and the chamber was opened. We evaluated the blood sugar of each rats prior to administration of single dose of alloxan monohydrate intraperitoneal at the dosage of 140 mg/kg by dissolving in normal saline. Acute hypoglycemia is reported by administration of alloxan so we provided the rats with 10% dextrose solution to avoid mortality percent. The fasting sugar level of rat’s blood were measured by commercially available glucometer (OnCall ^®^ Ez II; SN 303S0014E09) at 3rd, 7th, and 10th day post administration of alloxan injection. We selected the rats exhibiting fasting blood glucose level ≥ 250 mg/dL while kept on normal feed as shown in Table 1 to include in this research project.

### 2.7. Animals Grouping and Dose Administration

Thirty rats weighting 160–170 g were assigned into five groups (*n* = 06) as follows: (i) negative control group, control + vehicle; (ii) positive control group, CS and alloxan-induced diabetes; (iii) FM group, CS and alloxan-induced diabetes treated with FM extract; (iv) LM group, CS and alloxan-induced diabetes treated with LM extract and WPM group; CS and alloxan-induced diabetes treated with WPM extract. The ambient room temperature (24 ± 2 °C), light (12 h light/dark cycle), and humidity (62 ± 5%) were provided to all the animals. We administered the dose of *P. stewartii* Hf. plant extracts at 300 mg/kg body weight once a day in rats for 4 weeks with reference to guidelines of basic and clinical pharmacology and toxicology policies for clinical and experimental studies. Rats were dissected after the end of the trail period.

### 2.8. Blood and Organs Collection

Clot activator tubes were used for collection of blood samples and centrifuged for 20 min in centrifuge machine (makeup from China) having revolution power 1010× *g*. Eppendorf tubes were used for collection of serum and kept in a −20 °C biomedical freezer (Sanyo Biomedical Freezer, Japan) until further analysis. Collected tissue samples of liver and kidney were weighted and kept in 10% formalin buffered solution for 1 week. For histopathological analysis, tissues section were prior examined under a compound microscope (Model; IM–910, IRMEO Gmbh & Company, Lütjensee, Germany) outfitted with monocular camera (Model; TOUPCAM, Touptek Photonics Company., Ltd.; Zhejiang, China). On the basis of individual pathologic lesions in each section, the organ was stained with the H&E staining and followed the guidelines for preparation of histopathological slides as mentioned in [26]. Figure 1A shows the liver and kidney of a randomly selected representative rat from each group.

### 2.9. Biochemistry Study

Calorimetric method was used along with spectrophotometer (Model; Thermo Scientific Multiskan GO™ SkanIt software 4.1) by following company’s guideline for serological studies of liver, kidney, and oxidative stress markers. Stored serum was then thawed and evaluated for ALT, ALP, AST, HDL, LDL, creatinine, uric acid, bilirubin, globulin, albumin, total protein, and blood urea through biochemical kits commercially available through following their protocol leaflets. The oxidant and antioxidant values were evaluated by TOS, TAC, MDA, arylesterase, PON, and GSH levels through calorimetric method using a spectrophotometer following the procedure previously reported [27].

### 2.10. Statistical Analysis

Effect of *P. stewartii* Hf. treatment was studied through one-way ANOVA of statistical analysis along with post hoc evaluation by DMR test. Histopathological studies were performed through statistical analysis of Kruskal–Wallis test to establish effects of P. stewartii plant extracts. Data were illustrated as mean ± SE with 5% level of significance using SPSS software (21.0).

## 3. Results

### 3.1. Extraction Yield

The highest yield was obtained as 11.91%, 8.97%, 7.21% for FM, LM, and WPM respectively, where independent variables (200 mL, 8 h, 150 rpm) were set to study the response. Our results are in agreement with previous data that MPs of Phlomis species recorded significant yield in methanol [28]. Methanol has been identified as the most effective solvent for extraction of phytochemical compounds from MPs [29].

### 3.2. LC-ESI-MS Chromatogram and Fragmentation Pattern

*P. stewartii* Hf. plant was investigated for phenolic compounds by LC-ESI-MS. The LC-ESI-MS analysis of *P. stewartii* Hf. extracts showed the presence of various phytochemicals such as querctin, sinapinic acid, and vinallic acid as shown in Figure 2A. The identification of different compounds was made by comparison of retention time with standard investigation with same conditions and solvent. The ESI-MS spectra of the analyzed samples were compared with the standard and data from the literature. Figure 2B shows the mass spectrogram of sinapininc acid, and Figure 2C shows the fragmentation pattern of sinapininc acid. Table 2 shows the mass spectrum detail fragmentation peaks of sinapininc, vanillic, and querctin acids and their molar masses.

### 3.3. Cytotoxicity Assay

In this study, growth inhibitory potential of leaves, flower, and whole plant of *P. stewartii* Hf. plant in methanol was tested for cytotoxicity against HepG2 cell line. All these constituents were tested under comparable condition at different concentrations (12.5, 25, 50, 100, 200 µg/mL). All these components showed cytotoxicity against HepG2 cell line.

At a concentration of 200 µg/mL, the maximum cytotoxicity activity was found in LM, FM, and WPM which showed 46% in Figure 3A, 58% in Figure 3B, and 59% in Figure 3C growth inhibition against HepG2 respectively. Moreover results showed that cytotoxic activity of these extracts against HepG2 was recorded in the order LM > FM > WPM. The ¼ of IC_50_ value of LM, FM, and WPM were respectively, 187 > 280 > 312 µg/mL against HepG2 as shown in Figure 3D. It has been reported that a wide range of phenolic and polyphenolic components from *Phlomis* plant have been found to be effective cytotoxic agents such as kaempferol 3-*O*-glucoside, naringenin, verbascoside (acetoside), chlorogenic acids, and forsythoside B. Our results are in agreement with the previous studies showing that leaf extracts of medicinal plants have also been shown to inhibit the growth of human cancer cell [30].

### 3.4. Alteration in Hepatic Dysfunction Restoration

As expected, CS and alloxan-induced diabetes showed significant (*p* < 0.05) rise in ALT (136.4 µ/L), AST (303.2 µ/L), ALP (1093 µ/L), bilirubin (2.0 mg/dL), and LDL (123.2 mg/dL) at serum level in the positive control groups as compared to the negative control groups. Administration of *P. stewartii* Hf. plant improved the metabolic disturbance through lowering the levels of ALT, AST, ALP, bilirubin, LDL and increasing HDL. The administration of *P. stewartii* Hf. significantly (*p* < 0.05) improved the level of ALT (71.1 µ/L) and AST (239 µ/L and (246 µ/L)) in FM and WPM groups as compared to the positive control groups. ALP (862.3 µ/L) and LDL (89.6 mg/dL) showed significant (*p* < 0.05) differences in LM group as compared to the positive groups. The levels of AST and HDL showed significant (*p* < 0.05) improvement as compared to the negative control groups with treated groups of *P. stewartii* Hf. plant. Figure 4 shows that in all the treated groups the levels of total bilirubin are significantly (*p* < 0.05) different from the positive control group.

### 3.5. Improvement in Alterations in Hepatocellular Architecture

The metabolic disturbance generated through CS and alloxan-induced diabetes affected the liver strongly in positive control groups as compared to negative control and treated groups. The histopathological examination of negative control group (Left panel) indicated no fat accumulation, normal hepatic triad, and hepatocytes.

The histopathological analysis of positive control (middle panel) exhibited the peri-vascular, cytoplasmic vacuolation, and significant (*p* < 0.05) increase in portal cell infiltration, distorted hepatic triad, accumulation of fat, pyknotic and eccentric nuclei in hepatocytes. LM, FM, and WPM groups showed ameliorative effects of *P. stewartii* Hf. plant CS and alloxan-induced diabetes variations in liver tissue as shown in Figure 1B. The significance levels of histopathological analysis for liver tissues is mentioned in Table 3.

### 3.6. P. stewartii Hf. Alleviates Renal Damage

CS and alloxan-induced diabetes affected the renal system and significantly (*p* < 0.05) increased the creatinine (3.1 nmol/L) level in blood serum, urea (52.1 nmol/L), total protein (32.2 g/dL), and uric acid (7.2 mg/dL) in the positive control group. The administration of *P. stewartii* Hf. plant significantly (*p* < 0.05) improved the uric acid, creatinine, blood urea, and total protein levels as compared to the positive control groups as shown in Figure 5. The histological images of damaged nephrons also suggest the association of renal system exaggerated by CS and alloxan-induced diabetes in the positive control group. Histopathological study of the negative control group (left panel) demonstrated normal structures of proximal convoluted tubules and Bowman’s capsule. The positive control (middle panel) illustrated abnormal glomeruli having increased spaces at Bowman capsular regions. The *P. stewartii* Hf. groups (right panel) for histological analysis explained significant (*p* < 0.05) improvements in renal architecture in comparison to the positive control group. Figure 1C shows that administration of *P. stewartii* Hf. plant restored serum biomarker levels and ameliorated renal architecture as examined by histological inquiry. The mean ± SE of histopathological analysis for kidney tissues is given in Table 4.

### 3.7. Antioxidant Properties

Oxidative stress markers are considered as potent indicators for evaluating the homeostasis. Significant (*p* < 0.05) increase in TOS (148.8 µmol/L of H_2_O_2_ eq.), MDA (3.0 nmol/L) levels, and decrease in TAC (0.9 µmol/L of H_2_O_2_ eq.), PON (240.7 u/min/mL), arylesterase (68.6 Ku/min), and GSH (0.2 nmol per mg) levels were observed in the positive control PC groups. The administration of *P. stewartii* Hf. plant significantly (*p* < 0.05) decreased TOS and MDA levels, whereas TAC, arylesterase, and PON levels improved significantly (*p* < 0.05) representing anti-oxidant potential as compared to the positive control groups.

## 4. Discussion

Numerous bioactive components present in medicinal plants (MPs) with promising potential and antioxidant effect against human diseases through lowering oxidative damage have been reported over the past years. LC-ESI-MS analysis result shows the plant extract mainly possess three phenolics including sinapinic acid, vanilic acid, and quercetin in good quantities. Our results are in agreement with the reported data that plants of phlomis genus have different bioactive constituents [31,32]. In similar line, pervious study of P. stewartii Hf. showed the presence of various chemical constituents. It has been reported that presence of vanilic acid in MPs as affordable and novel neuroprotective agent against cerebrovascular insufficiency states, vascular dementia, hepatic and renal diseases [13]. Some recent studies reported that aside from the antioxidant potential of the majority of components present in vanilic acid class such as phenolic compounds, they have been mentioned to perform different activities at molecular levels which makes it an effective candidate for more detailed analysis as well. In our study, the hepatoprotective effects of *P. stewartii* Hf. plant extract are due to the presence of vanilic acid as shown in Figure 4. Quercetin has been reported to possess ketocarbonyl group in its molecule and presence of oxygen atom in the first carbon atom is basic and can generate salts with acids. Its molecular structure has four different groups such as dihydroxy group between A ring, o-dihydroxy group B, C ring C2, 4-carbonyl and C3 double bond [33]. Previous studies suggested that quercetin possess anti-microbial, anti-diabetic, and anti-oxidant properties due to the presence of double bond and phenolic hydroxyl group [34]. Querctin rapidly metabolized and excreted from the body without affecting the vital organs and possesses short half-life in blood [35]. It plays important role in maintaining the redox balance of the body and also improved the expression of glutathione peroxide dismutase (GSH), catalase (CAT), superoxide dismutase (SOD), and used as a tool to cure the different age-associated disorders [36]. In our study we found the promising antioxidants potential of quercetin present in *P. stewartii* Hf. plant extracts through measuring the oxidative stress markers as shown in Figure 6. Sinapinic acid, a hydroxycinnamic acid derivative, has antioxidant potential which is responsible for the decline of oxidative stress and improvement of antioxidant status in plasma and tissues of living body [37]. In addition, through preservation of mitochondrial function these sinapinic acids attenuates cardiac hypertrophic responses [38]. Sinapinic acid possess effective potential against hyperglycemia-mediated oxidative stress in experimental type 2 diabetes mellitus (DbT2) in rats [39]. In our study, *P. stewartia* Hf. Plant extract showed effective anti-diabetic potential which might be attributed to the presence of sinapinic acid. In similar lines, our results are in agreement with previous reports that MPs of *Phlomis* species have a wide range of bioactive constituent which are responsible for antioxidants, anti-diabetic, anti-inflammatory, anti-microbial effects [40].

Liver, which is a vital metabolic organ, is involved in various biochemical processes of the body in maintaining homeostasis. Liver diseases as its comorbidities such as hypertension, obesity, oxidative stress, insulin resistance, exercise intolerance, diabetes mellitus favor the proinflammatory markers [41]. If the pathogens accumulate and translocation process hampers, viruses and bacteria stimulate the production of pathogen-associated molecular patterns (PAMPs) in general circulation which in the liver causes chronic diseases. The expression of Toll-like and pattern recognition receptors (TLPRs) mediated through PAMPs activates inflammatory processes which results in necrosis, fibrosis, and apoptosis of liver cells [42]. The metabolic disorders such as glucose intolerance, insulin resistance dyslipidemia, hypertension, DM, and obesity lead to metabolic syndrome, that is a major factor in developing non-alcoholic fatty liver diseases (NAFLD). NAFLD appear from simple steatosis of liver cells which ultimately lead to nonalcoholic steatohepatitis (NASH). NASH results from metabolic syndrome, and causes hepatocellular carcinoma [43]. The cytotoxic activity studied through MTT assay of these extract against HepG2 was recorded in the order LM > WPM > FM. The ¼ of IC_50_ value of LM, WPM, and FM were respectively, 187 > 280 > 312 µg/mL against HepG2.

The ameliorative effects of *P. stewartii* Hf. plant might be attributed to its antilipogenic properties. *P. stewartii* Hf. plant have contributed their role for regulating liver enzyme to minimize oxidative stress. Lipid homeostasis maintains equilibrium through balance between lipid production and utilization. In metabolic syndromes, abnormalities occur in lipogenic and lipolytic pathway. As previous literature suggested, increased load of triglycerides and cholesterol mediate metabolic syndrome toward DM [44,45]. The beneficial properties of *P. stewartii* Hf. plant due to its bioactive components such as flavone glycoside, phenylethanoid, coumarins, and tannin [24,46] evoke the following mechanisms: (1) reduced the toxin levels generated by food components to minimize the reactive oxygen species; (2) strengthens host innate response and adaptive immunity; (3) minimizing bacterial derived hepato-toxins especially acetaldehyde, ethanol, volatile fatty acids, (4) maintains appetite and satiety. The objective of the present study was to comprehensively explore the potential of indigenous *P. stewartii* Hf. plant methanolic extract for cytotoxic, antidiabetic, hepatoprotective, nephroprotective, and antioxidative properties. It has been reported that metabolic syndromes increase LPS generation which hamper glucose metabolism that develops hyperglycemia, hypercholesterolemia, hepatosteatosis, and DM [47]. The elevated levels of AST, ALT, ALP, blood urea, creatinine, uric acid, total proteins along with distorted histopathology of liver and kidneys have also been reported in previous studies of DM mediated through metabolic syndromes [48,49]. The liver biomarkers as shown in Figure 3 are considered as baseline parameters to declare the hepatic impairment, their prolonged abnormal levels in body leads toward inflammation then necrosis, steatohepatitis, and ultimately hepatocellular carcinoma [50]. The treatment of metabolic syndrome model with *P. stewartii* Hf. indicates ameliorative effects in improving the lipid profile through lowering the metabolic stress. These biochemical parameters are thought as baseline parameters in declaring metabolic syndrome. In our study, significant (*p* < 0.05) increase in ALT, AST, ALP, blood urea, creatinine, uric acid, bilirubin, and total proteins in positive control groups strongly suggested the models of CS and alloxan-induced diabetes.

The function of liver and kidney in mediating the pathogenesis of DM occurred through obesity, OS, and metabolic syndromes resulting in increased blood flow through hepatic-portal system which leads to necrosis and fibrosis of digestive organs, particularly liver along with accessory gland pancreas [51]. Uremic toxins associated with metabolic diseases especially obesity and DM contribute to the progression of renal impairment [52]. The compromised excretion and emulsification of fats results in urates deposition in nephrons leading to renal damage along with DM [53]. The administration of *P. stewartii* Hf. plant strengths the immune system and improves nitrogen cycling. The metabolism affects the short chain fatty acids production which is involved in renal blood flow through olfactory receptor, a G-protein coupled receptor present in renal juxtraglomerular region, affects renal secretion.

The association of liver and kidney in mitigating endotoxemia through CS and alloxan-induced diabetes is presented [54]. In our study, the CS and alloxan-induced diabetes significantly increased endotoxin in blood urea, creatinine, uric acid, and total protein levels in the positive control group. The renal biomarkers in Figure 4 are considered as diagnostic parameters to assess the renal function of particular living body. The abnormal levels of renal biomarkers push the body toward renal impairment eventually leading to renal failure [55]. The extracts of *P. stewartii* Hf. showed their potential to minimize the metabolic endotoxemia and uremic toxins in mitigating the disease load and helping the body in improving renal health.

In metabolic syndromes, irregularities are frequently experienced in portal triad and cytoplasm of hexagonal hepatic lobules in liver and damaged nephron in kidneys [54]. These abnormalities along with fat accumulation, necrosis, and fibrosis have been showed in our studied model. The administration of *P. stewartii* Hf. plant lowers lipid peroxidation at hepatocellular levels through subsiding inflammation, improving hepatic portal triad, as shown in Figure 1B, minimizing fat accumulation, restoring damaged nephron and urates deposition as shown in Figure 1C.

OS is linked with irregular secretions of adipokines that facilitates metabolic syndrome [56,57]. Metabolic oxidative stress is caused through high cholesterol, triglycerides, and LDL oxidation [58]. The increased TOS and MDA levels showed peroxidation of fatty acids [47]. The anti-oxidative effects of *P. stewartii* Hf. plant in the current study might be due to ROS scavenging potential through inhibition of nitric oxide production and caveolin signaling, modulation of superoxide dismutase and glutathione peroxidase genes expression. The oxidant and antioxidant parameters in Figure 5 are considered as a potent system to manage the balance between oxidants and anti-oxidants levels in the body. The free radicals of oxygen, nitrogen, and sulfur are produced in the body through end products of metabolic phenomena. The living body is protected with anti-oxidant defense system, which protects from damage results of oxidative stress and cell injury [59]. We found many parameters in our study which show the potential of *P. stewartii* Hf. plant in combating the oxidative stress significantly.

The presence of different bioactive compounds in *P. stewartii* Hf. such as cardiac glycoside, phlobatannins, quinone, phenylethanoids, nortriterpenoids, and oleanolic acids have potent pharmacological properties in mitigating the OS-related metabolic syndromes. The cytotoxic and antidiabetic potential of *P. stewartii* Hf. [60] is due to antioxidative properties. Moreover, reducing the metabolism of carbohydrates by inhibiting the α-glucosidase and α-amylase enzyme, the flavonoids and phenolic components may exert an antidiabetic effect by increasing the insulin secretion, decreasing the intestinal carbohydrates absorption, increasing β-cell functions and antioxidants effects. The results from our study are parallel with the findings of previous studies that methanol extracts of medicinal plants possess potent antioxidant properties and bioactive constituents which are considered as important in the treatment of DM [50,61].

## 5. Conclusions

The cytotoxic effects of *P. sterwartii* Hf. plant was studied through MTT assay which revealed its potential to inhibit growth against HepG2 cell lines. Treatment with *P. stewartii* Hf. methanol extract at 300 mg/kg body weight ameliorates the abnormalities generated in the rat model of CS and alloxan-induced diabetes. The results from our study revealed that extracts of *P. stewartii* Hf. possess anticancer, antioxidant, hepatoprotective, and nephroprotective potential and serve as potential therapeutic agents in combating DM. Further studies should be designed to explore the mechanistic approaches of the bioactive constituents in *P. stewartii* Hf. plant.

## Figures and Tables

**Figure 1 metabolites-12-01133-f001:**
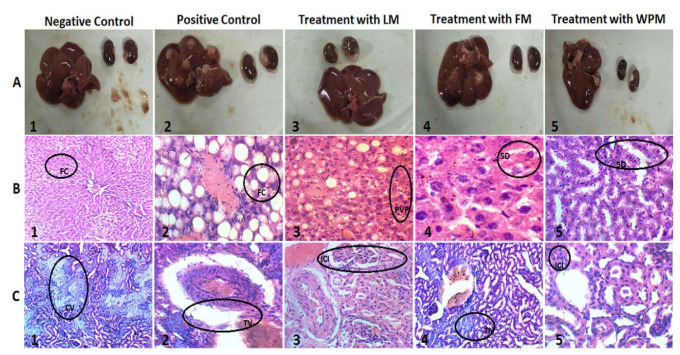
Effect of *P. stewartii* Hf. plant extract on; (**A**) liver morphology, (**B**) liver histology and (**C**) renal histology.

**Figure 2 metabolites-12-01133-f002:**
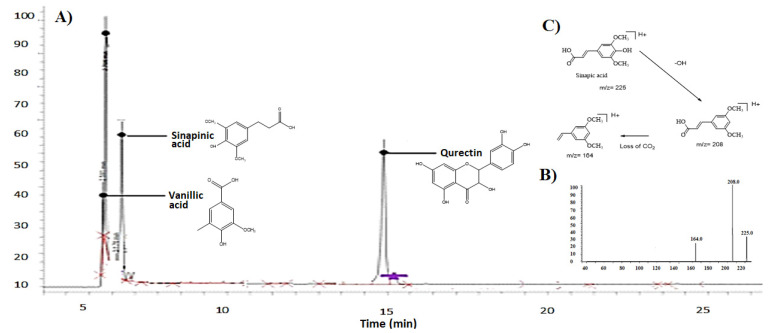
(**A**) HPLC chromatogram of *P. stewartii* Hf. plant extract, (**B**) mass spectrogram of sinapinic acid and (**C**) fragmentation pattern of sinapinic acid.

**Figure 3 metabolites-12-01133-f003:**
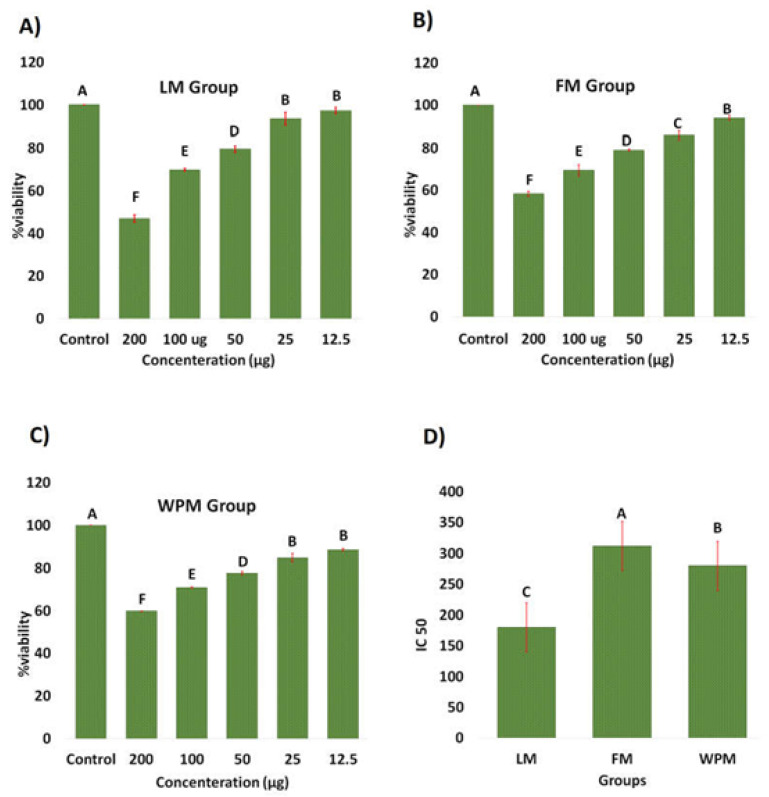
Effect of; (**A**) leaf extract, (**B**) flower extract, and (**C**) whole plant extract on the viability of HepG2 cells line and (**D**) their cytotoxicity IC50 values. The different alphabet at chart bars showing significant difference.

**Figure 4 metabolites-12-01133-f004:**
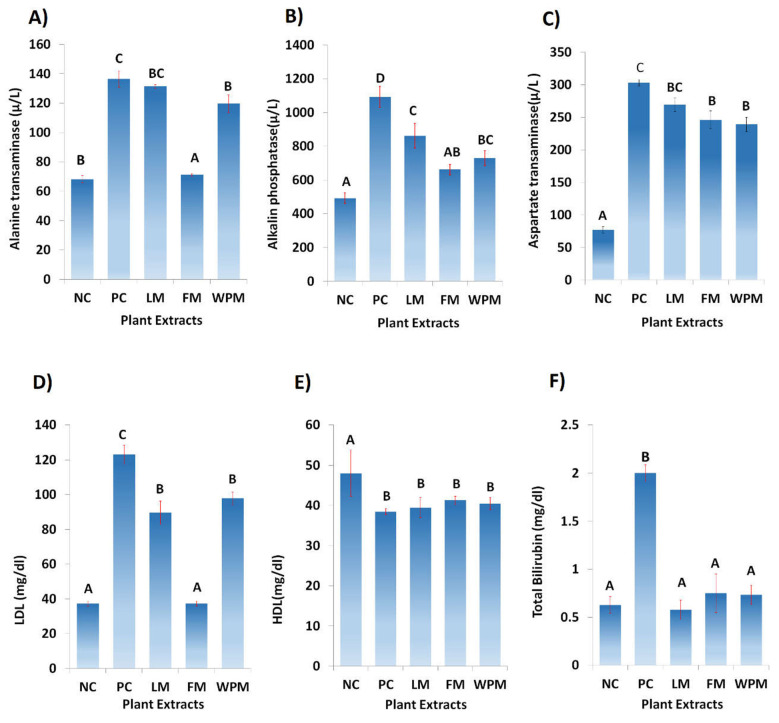
Effect of leaf, flower, and whole plant extract in contrast to negative and positive control (NC & PC) on serum lipid profile; (**A**) alanine transaminase, (**B**) alkaline phosphatase, (**C**) aspartate transaminase, (**D**) low density lipoprotein, (**E**) high density lipoprotein, and (**F**) total bilirubin. The different alphabet at chart bars showing significant difference.

**Figure 5 metabolites-12-01133-f005:**
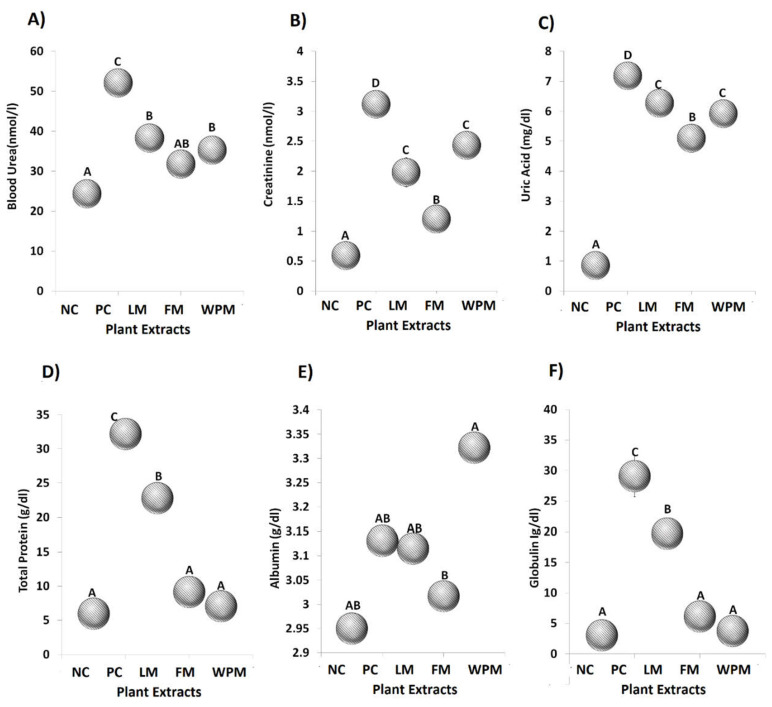
Effect of leaf, flower, and whole plant extract in contrast to negative and positive control (NC and PC) on renal function biomarkers; (**A**) blood urea, (**B**) creatinine, (**C**) uric acid, (**D**) total protein, (**E**) albumin, and (**F**) globulin. The different alphabet at chart bars showing significant difference.

**Figure 6 metabolites-12-01133-f006:**
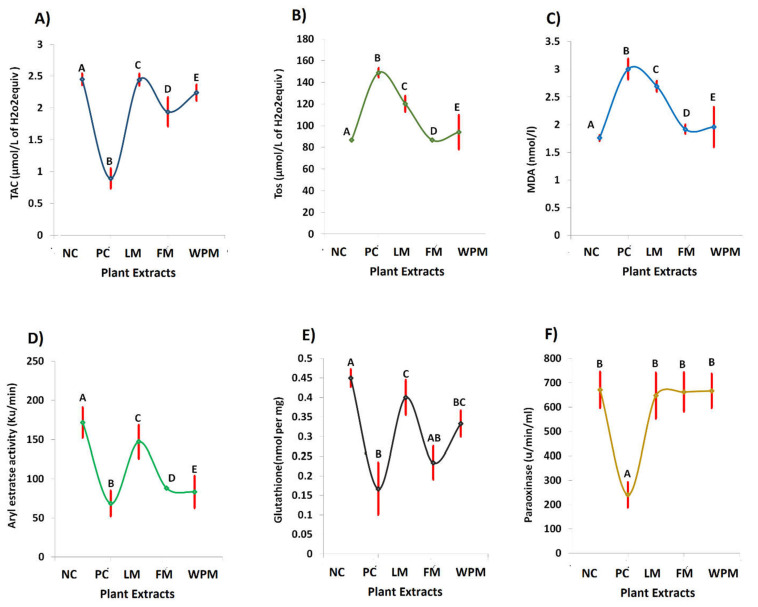
Effect of leaf, flower, and whole plant extract in contrast to negative and positive control (NC and PC) group on ameliorate cigarette smoke and alloxan-induced diabetes biomarkers; (**A**) total antioxidant capacity, (**B**) total oxidant status, (**C**) malondialdehyde, (**D**) arylesterase, (**E**) paraoxinase, and (**F**) glutathione. The different alphabet at chart bars showing significant difference.

**Table 1 metabolites-12-01133-t001:** Normal diet composition for rats.

	Normal Diet
Sucrose	01
Ash	5%
Fat	6%
Crude protein	18%
Crude fibre	5.5
Nitrogen free extract (NFE)	64.5%

**Table 2 metabolites-12-01133-t002:** Chemical compounds in *P. stewatrii* Hf. extracts determine by LC-ESI-MS.

Major LC/MS *m*/*z* (Intensity)		Molecular for-Molar Mass Mula		Name of Identified Compound
1.73	169 (90), 135 (45).	C8H8O4	168	Vanillic acid
4.00	303 (100) 285 (20), 243.082, 178.998, 151.004	C15H10O	302	Querctin
1.85	225 (30), 208 (100), 164 (20).	C11H12O5	224	Sinapinic acid

**Table 3 metabolites-12-01133-t003:** Histological parameters of liver in cigarette smoke and alloxan-induced diabetes.

	Positive Control	Negative Control	Treatment Group	Kruskal Wall Test for Histology Asymptotic Significance Level (*p* < 0.05)
Sinusoidal dilatation	00.25 ± 00.11	1.44 ± 00.12	0.83 ± 0.02	0.005
Focal necrosis	00.16 ± 00.09	1.51 ± 00.08	0.91 ± 0.15	0.001
Pyknotic nuclei	00.53 ± 00.01	1.53 ± 00.01	0.92 ± 0.12	0.002
Lobular necrosis (Central)	00.16 ± 00.08	1.81 ± 00.0	0.92 ± 0.18	0.001
Eccentric nuclei at hepatocytes	00.24 ± 00.07	2.01 ± 00.17	1.0 ± 0.22	0.001
Cytoplasmic vacuolation of tubular epithelium	00.33 ± 0.09	1.51 ± 0.02	00.76 ± 0.11	0.01

**Table 4 metabolites-12-01133-t004:** Histological parameters of kidney in cigarette smoke and alloxan-induced diabetes are associated with renal damage.

	Positive Control	Negative Control	Treatment Group	Kruskal Wall Test for Histology Asymptotic Significance Level (*p* < 0.05)
Renal Tubular thickening	00.166 ± 0.10	1.51 ± 00.006	1.07 ± 00.10	0.002
Infiltration of Interstitial	00.165 ± 0.10	1.76 ± 00.07	00.90 ± 00.15	0.004
Cytoplasmic vacuolation at renal tubular epithelium	00.24 ± 0.07	1.92 ± 00.08	00.84 ± 00.15	0.011
Renal Tubular necrosis	00.25 ± 0.07	1.59 ± 00.22	1.0 ± 00.11	0.008

## Data Availability

Data is contained within the article.

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
