# Peer review of "Anti-Diabetic and Cytotoxic Evaluation of Phlomis stewartii Plant Phytochemicals on Cigarette Smoke Inhalation and Alloxan-Induced Diabetes in Wistar Rats"

_metabolites, 2022, doi:10.3390/metabo12111133_

Round 1

Reviewer 1 Report

Review on the manuscript.

Title: Anti-diabetic and cytotoxic evaluation of Phlomis stewartii 2 plant phytochemicals on cigarette smoke inhalation and 3 alloxan induced diabetes in Wistar rats

In the present study,the author has made  methanol extracts from leaves, flower and whole plant of Phlomis stewartii were analyzed using LC-ESI-MS and investigated for their cytotoxic properties against HepG2 cell line and CS  alloxan induced diabetes in Wistar albino rats model. Rats were kept in aerated cage for eight weeks induced diabetes mellitus (DM) by administration of alloxan@140 mg/kg.The cytotoxic activity of extracts against HepG2 was recorded The IC50 values were recorded. According to the results of this study, Phlomis Stewartii methanol extracts has shown good antioxidant, anticancer activity and worked well to recover the tested clinical parameters in CS/alloxan induced diabetes animals.

The manuscript was well written without any ambiguity.

In my opinion the manuscript can be accepted for publication with the following clarifications.

In the abstract it says plant extracts were analysed by LC-ESI-MS whereas in the result and discussion it talks about LC-ESI-MS/MS. Discussion of the Analysis by LC-ESI-MS/MS, was not satisfactory.  Some more insights required.

Line No. 356 & 357: Previous studies suggested that quercetin possess anti-microbial, anti-diabetic and anti-oxidant properties due to presence of double bond and phenolic group. Which double bond is responsible? Do you mean all phenolic compounds will possess anti-microbial, anti-diabetic and anti-oxidant properties?

Sinapinic acid 364 is responsible for decline of oxidative stress and improvement of antioxidant status in 365 plasma and tissues of living body.  Jusify

Line No: 459:  Overall, cytotoxic and antidiabetic potential of Phlomis stewartii is due to antioxida tive properties.

Do you mean all antioxidants will have cytotoxic and ant diabetic activity. Justify

Line No. 471 to 474: The authors have concluded  from their investigation that different bioactive constituents of Phlomis stewartii such as cardiac glycoside, phlobatannins, quinone, phenylethanoids, 28-nortriterpenoids, and oleanolic acids have potent pharmacological properties that are responsible for beneficial effects in mitigating the OS related metabolic syndromes.

There is no mention of these constituents in the results and discussion or in LC-MS/MS section.  Then how it could be concluded like this.

Author Response

Dear Sir

The manuscript entitled “Anti-diabetic and cytotoxic evaluation of Phlomis stewartii plant phytochemicals on cigarette smoke inhalation and alloxan induced diabetes in Wistar rats” has been revised according to the comments, detail of response to reviewer’s comments are given below. I hope you will find that we have justified all the comments accordingly.

Best Regards

Prof. Dr. Syed Ali Raza Naqvi

Response to comments – Reviewer 1

Title: Anti-diabetic and cytotoxic evaluation of Phlomis stewartii 2 plant phytochemicals on cigarette smoke inhalation and 3 alloxan induced diabetes in Wistar rats

In the present study,the author has made  methanol extracts from leaves, flower and whole plant of Phlomis stewartii were analyzed using LC-ESI-MS and investigated for their cytotoxic properties against HepG2 cell line and CS  alloxan induced diabetes in Wistar albino rats model. Rats were kept in aerated cage for eight weeks induced diabetes mellitus (DM) by administration of alloxan@140 mg/kg.The cytotoxic activity of extracts against HepG2 was recorded The IC50 values were recorded. According to the results of this study, Phlomis Stewartii methanol extracts has shown good antioxidant, anticancer activity and worked well to recover the tested clinical parameters in CS/alloxan induced diabetes animals.

The manuscript was well written without any ambiguity.

In my opinion the manuscript can be accepted for publication with the following clarifications.

  1. In the abstract it says plant extracts were analysed by LC-ESI-MS whereas in the result and discussion it talks about LC-ESI-MS/MS. Discussion of the Analysis by LC-ESI-MS/MS, was not satisfactory. Some more insights required.

Response

Thanks for withdrawing our attention to mistakenly written LC-ESI-MS/MS, it is LC-ESI-MS. We have fixed the issue and necessary discussion have also been added.

  1. Line No. 356 & 357: Previous studies suggested that quercetin possess anti-microbial, anti-diabetic and anti-oxidant properties due to presence of double bond and phenolic group. Which double bond is responsible? Do you mean all phenolic compounds will possess anti-microbial, anti-diabetic and anti-oxidant properties?

Response

It is general discussion repported in previous published studies that a particular double bond may invovle in biological activites, however, in our view it is synergic effect and not a particular double bond do this job. Moreover we have added more refrences relevant to phenolic compounds having anti-microbial, anti-diabetic and anti-oxidant properties.

  1. Sinapinic acid is responsible for decline of oxidative stress and improvement of antioxidant status in plasma and tissues of living body. Justify

Response

This is well documented that sinapinic acid, a hydroxycinnamic acid derivative, is responsible for decline of oxidative stress and improvement of antioxidant status in plasma and tissues of living body, to justify we have added reference for justification.  

  1. Line No: 459: Overall, cytotoxic and antidiabetic potential of Phlomis stewartii is due to antioxidative properties. Do you mean all antioxidants will have cytotoxic and ant diabetic activity. Justify

Response

The presence of different bioactive compounds present in P. stewartii such as cardiac glycoside, phlobatannins, quinone, phenylethanoids, 28-nortriterpenoids, and oleanolic acids have potent pharmacological properties in mitigating the OS related metabolic syndromes. However, in herbal medicinal point of view, it is synergic effect of combination of different compounds. This is not completely related with antioxidants.

  1. Line No. 471 to 474: The authors have concluded from their investigation that different bioactive constituents of Phlomis stewartii such as cardiac glycoside, phlobatannins, quinone, phenylethanoids, 28-nortriterpenoids, and oleanolic acids have potent pharmacological properties that are responsible for beneficial effects in mitigating the OS related metabolic syndromes. There is no mention of these constituents in the results and discussion or in LC-MS/MS section.  Then how it could be concluded like this.

Response

We have improved the results and discussion section by discussing these compounds.

Reviewer 2 Report

Dear authors,

I would like to thank you very much for this very comprehensive manuscript. If it is possible, It would be appropriate to increase the quality of figure 2.

Best regards.

Author Response

Response to comments – Reviewer 2

I would like to thank you very much for this very comprehensive manuscript. If it is possible, It would be appropriate to increase the quality of figure 2.

Response

Dear reviewer thanks for appreciation, we will place the high quality Figure 2 in the final version.

Reviewer 3 Report

Comments to Authors

The manuscript entitled “Anti-diabetic and cytotoxic evaluation of Phlomis stewartii plant phytochemicals on cigarette smoke inhalation and alloxan induced diabetes in Wistar rats” is a good piece of work. The authors have done good job by working on the bio potential of phytochemical extracts of different parts of Phlomis stewartii plant using three different solvents. The introduction, material & methods, sections fully support the theme of work. The results and discussion part is comprehensive and arguments-based. LC/MS data is also very informative and the authors have well described the relation of different phytochemicals in antidiabetic and cytotoxic activities of the extracts. However, there are some small mistakes throughout. They are as follows;

1. The authors are encouraged to concise the caption of Figure-1 and the other figures throughout.

2. The authors are encouraged to include the botanical authority of Phlomis stewartii once as per IPNI index/The plant list database.

3. There are formatting mistakes at many places. For instance, in first reference, the list of authors is not full. The authors are encouraged to look for similar mistakes throughout.

4. At some points there are typo graphic and spellings mistakes, please go through and fix them.

5. The authors are encouraged to add few latest references in the introduction part related to this work.

After incorporating the above minor changes, I would like to recommend this manuscript for publication in this journal.

Author Response

Response to comments – Reviewer 3

The manuscript entitled “Anti-diabetic and cytotoxic evaluation of Phlomis stewartii plant phytochemicals on cigarette smoke inhalation and alloxan induced diabetes in Wistar rats” is a good piece of work. The authors have done good job by working on the bio potential of phytochemical extracts of different parts of Phlomis stewartii plant using three different solvents. The introduction, material & methods, sections fully support the theme of work. The results and discussion part is comprehensive and arguments-based. LC/MS data is also very informative and the authors have well described the relation of different phytochemicals in antidiabetic and cytotoxic activities of the extracts. However, there are some small mistakes throughout. They are as follows;

  1. The authors are encouraged to concise the caption of Figure-1 and the other figures throughout.

Response

Caption of Figure-1 and the other figures throughout the manuscript have been concised.

  1. The authors are encouraged to include the botanical authority of Phlomis stewartii once as per IPNI index/The plant list database.

Response

The botanical authority has been included in the manuscript.

  1. There are formatting mistakes at many places. For instance, in first reference, the list of authors is not full. The authors are encouraged to look for similar mistakes throughout.

Response

The references have been crosschecked and formatted into journal style.

  1. At some points there are typo graphic and spellings mistakes, please go through and fix them.

Response

After proper reading we have corrected the typographic and spelling mistakes.

  1. The authors are encouraged to add few latest references in the introduction part related to this work.

Response

We have added few latest references in introduction part related to this work.

Reviewer 4 Report

The comments are listed below:

Lines 78, 83, 86, 91, 97, 109, 135, 170, 191, 195, 196, 199, 202, 203, 204, 218, 231, 232, 241, 268, 270, 275, 279, 296, 300, 311, 313, 317, 330, 336, 354, 363, 369, 391, 392, 397, 403, 413, 426, 436, 442, 449, 457, 459, 472, 476, 479, Phlomis stewartiiP. stewartii

Line 138, 25Ul→ 25 µL

Lines 547 & 574, Phlomis stewartii Phlomis stewartii

Lines 103 & 164, 10g→ 10 g

Lines 98, 106, 107, 120, 127, 169,177, 30°C30 °C

Lines 138 & 139, hs→ h

Lines 257 & 389, IC50→ IC50

Lines 138, 147, 170, 303, 5mg→ 5 mg

Line 260, 3-O-glucoside3-O-glucoside

Line 329, 0.2nmol0.2 nmol

Lines 327 & 328, H2o2equivH2O2equiv

Author Response

Response to comments – Reviewer 4

  1. Lines 78, 83, 86, 91, 97, 109, 135, 170, 191, 195, 196, 199, 202, 203, 204, 218, 231, 232, 241, 268, 270, 275, 279, 296, 300, 311, 313, 317, 330, 336, 354, 363, 369, 391, 392, 397, 403, 413, 426, 436, 442, 449, 457, 459, 472, 476, 479, Phlomis stewartii→  stewartii

Line 138, 25Ul→ 25 µL

Lines 547 & 574, Phlomis stewartii→ Phlomis stewartii

Lines 103 & 164, 10g→ 10 g

Lines 98, 106, 107, 120, 127, 169,177, 30°C→ 30 °C

Lines 138 & 139, hs→ h

Lines 257 & 389, IC50→ IC50

Lines 138, 147, 170, 303, 5mg→ 5 mg

Line 260, 3-O-glucoside→ 3-O-glucoside

Line 329, 0.2nmol→ 0.2 nmol

Lines 327 & 328, H2o2equiv→ H2O2equiv

Response

Throughout the manuscript Phlomis stewartii was replaced with P. stewartii Hf. (botanical name). All these technical errors have been incorporated throughout the manuscript.